# An Integrated Approach to Assess the Potential of Forest Areas for Therapy Services

**Yonko Dodev** [1], **Miglena Zhiyanski** [1,*], **Maria Glushkova** [1], **Bilyana Borisova** [2], **Lidiya Semerdzhieva** [2], **Ivo Ihtimanski** [3], **Stelian Dimitrov** [4], **Stoyan Nedkov** [3], **Mariyana Nikolova** [3] and **Won-Sop Shin** [5]

1   Forest Research Institute, Bulgarian Academy of Sciences, 132 "St. Kliment Ohridski" Blvd., 1756 Sofia, Bulgaria; ionkododev@abv.bg (Y.D.); m_gluschkova@abv.bg (M.G.)
2   Landscape Ecology Department, Faculty of Geology and Geography, Sofia University "St. Kliment Ohridski", 15 "Tzar Osvoboditel" Blvd., 1504 Sofia, Bulgaria; billiana@gea.uni-sofia.bg (B.B.); l.nikolaeva@gea.uni-sofia.bg (L.S.)
3   National Institute of Geophysics, Geodesy and Geography, Bulgarian Academy of Sciences, Akad. G. Bonchev Str. Bl.3, 1113 Sofia, Bulgaria; ivo.ihtimanski@gmail.com (I.I.); snedkov@geophys.bas.bg or snedkov@abv.bg (S.N.); mn@bas.bg (M.N.)
4   Cartography and GIS Department, Faculty of Geology and Geography, Sofia University "St. Kliment Ohridski", 15 "Tzar Osvoboditel" Blvd., 1504 Sofia, Bulgaria; stelian@gea.uni-sofia.bg
5   Department of Forest Sciences, Chungbuk National University, 1 Chungdaero, Seowongu, Cheongju 28644, Korea; shinwon@chungbuk.ac.kr
*   Correspondence: miglena.zhiyanski@bas.bg

**Abstract:** The study considers forest therapy as a tool for diversification of forest management. An up-to-date integrated approach for assessing and mapping potential of forest areas which could provide conditions for forest therapy services is developed and tested. It is based on combining data from the traditional forest inventory in Bulgaria and other open databases with methods for integrated assessment and mapping of ecosystem services: 7 criteria groups and 22 indicators are proposed, rated on a 5-point scale. Overlay analysis is applied to generate a composite assessment for each forest unit. Using spatial statistics tools, territorial hot spots with potential for forest therapy are identified. The methodology was successfully tested in a pilot case-study region, Smolyan Municipality, but it is applicable at broader scale, regardless of the type and ownership of forests. This approach could be transferred to other countries as well after adapting to their geographical, geoecological and socio-cultural specifics and database available. It is a cost-effective and informative tool to support forest owners and managers to diversify forest welfare services focusing on insufficiently used forest recreation potential.

**Keywords:** forest therapy; forest recreation; integrated assessment and mapping; forest management; forest welfare services; cultural ecosystem services

## 1. Introduction

Forests provide a wide range of economic, ecological, and social benefits to humankind by contributing to the overall economy through employment, processing and trade of forest products and energy, providing services and investments in the forest sector [1]. They also include hosting and protection of sites and landscapes of high cultural, spiritual, or recreational value [1]. Maintaining and enhancing these cultural ecosystem services should be an integral part of any forest policy as the information on status and trends in socio-economic benefits are essential in evaluating progress towards sustainable forest management.

The economic benefits are usually measured in monetary terms [1], but while the intangible benefits of forests for human health and wellbeing are often proven their monetary measurement is difficult. These intangible benefits can vary considerably among countries depending on their level of development and traditions and bearing in mind that

these specifics are often measured in terms of area or proportion of forests used to provide various social services.

The economic, social, and ecological role of forests is of significant importance to the sustainable development of society and for improving the quality of life, especially in rural and remote areas. These functions are unique not only in a national but also in a global context. This emphasizes the need for forest resources to be professionally managed in a stable forest sector with broad public support and mutual respect and integration of the interests of all stakeholders.

Bulgarian forests and other wooded land cover 4.27 million hectares, representing approximately 38% of the country's land area [2]. They are considered a national asset and recognized as a part of the natural heritage and national identity. The current forest policy in Bulgaria follows the EU notions for sustainable and multifunctional forest management [3,4]. It is expressed in searching for an optimal balance in the use of the various forest ecosystem goods and services (provisioning, supporting, regulation & maintenance and cultural) [5] in order to meet the widest possible range of public needs and at the same time to prevent forest degradation and biodiversity loss. Yet while the use of basic forest provisioning, supporting and regulation & maintenance ecosystem services is statutory and management structures are familiar with them, this does not apply to the forest cultural ecosystem services. They refer to the use of forests for recreation, leisure and education and are identified as a priority in all main forest policy documents in Bulgaria (Forest Act, National Forest Strategy, National Strategy for Regional Development, ratified UN Convention for biological diversity, National Concept for Spatial Development, Protected Territories Act, Game Act etc.). Despite the formal recognition, the cultural ecosystem services are still insufficiently studied and no real measures (including legislation, services, facilities, management structures, experts, etc.) have been developed. In this regard, the comprehensive concept for Forest Welfare Services was recognized and recently its development was initiated [6]. It refers to cultural services for the population based on forests and forest resources which aim to improve the human health and wellbeing, including forest therapy, recreation, education, cultural and sport activities. Moreover, the failure to fully value spiritual/cultural aspects of trees may lead to mismanagement [7].

An essential element of the Forest Welfare Services is the Forest Therapy (Healing). Forest Therapy involves the use of natural forest elements to promote human health and enhancing human immunity by using natural forest elements. It is practiced for both prevention and treatment of diseases and to increase life expectancy. It may comprise a complex of diverse activities performed in forests—forest bathing, herbal therapy, aromatherapy, physical therapy, meditation, sun, and air baths, etc. [8–21]. The effect of forest therapy on human health is an important topic in modern multidisciplinary research [22–24] and the recent COVID-19 pandemic situation in the last years has provoked the research and public interest on the role of forests. At the policy level this crisis may also present opportunities for pro-environmental change [25,26]. Evidence of the healing effect of forests motivates further research to improve knowledge of forests as an environment for therapeutic recreation and tourism [27,28], including development of tools for precise spatial analysis and mapping. Upgrading scientific knowledge is important for the implementation of management of forest resources adequate to modern societal needs which requires an in-depth study of the possibilities of integrating forest therapy into forest management for maintaining well-being levels [29]. This also provides a new direction for analysis of the traditional forest inventory which does not assess the forest therapy potential of forest areas.

This study highlights an opportunity for the European forestry to renew and diversify the social recreational and health aspect of forest management through the adaptation of best practices—the Korean experience in the deployment of "forest therapy", to European geographical and socio-cultural conditions. The study uses available scientific knowledge in the identification and valuation of cultural ecosystem services to test an integrated approach to assess the potential for forest therapy in a representative forest region of Bulgaria. In this sense, we consider the study and its results as a new perspective on European

practices, which may be of serious interest to forest owners and forest management, as well as to those active in nature-based tourism and recreation.

The aim of this study is to develop and test an integrated methodology for assessment and mapping of potential of forest areas in Bulgaria to provide therapy services. The idea is based on integration of traditional data from the forest inventory in Bulgaria with up-to-date tools for mapping and assessment of cultural ecosystem services.

## 2. Materials and Methods

The case-study region of Smolyan municipality located in Rhodopes Mountains, Bulgaria (Figure 1) was chosen as a pilot for testing the approach developed. Forest lands occupy 65,650 ha, representing 76% of its total area [30]. The territorial scope of Smolyan Municipality includes a total area of 865 km$^2$, 65,650 ha total forests with 29,579 forest subdivisions. The forest areas are of great importance to the economy of the region, mainly as a source of construction and technological timber as well as for providing non-timber forest products, protecting biodiversity and giving options for recreation. Therefore, it is especially important targeted actions for multifunctional utilization of the forest territories in the region to be planned and implemented. As a "typical forest area" Smolyan case-study region provokes the increased interest of the local community, forest management authorities, and business entrepreneurs to determine opportunities and directions for further development of the municipality as regards to diversify forestry sector.

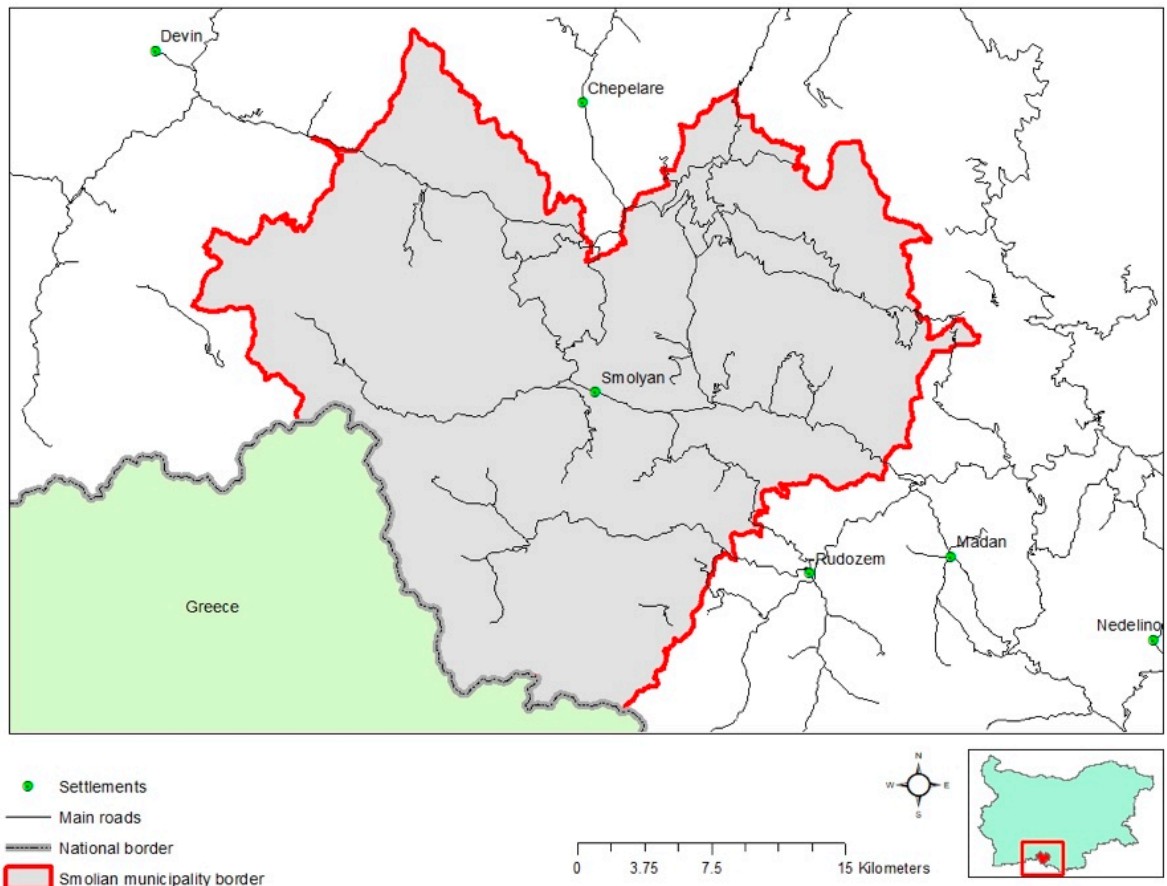

**Figure 1.** Case-study region: Smolyan municipality.

Smolyan Municipality has a high touristic potential based on the unique combination of beautiful and diverse nature, natural phenomena, excellent climatic conditions, mineral springs, millennial cultural and historical heritage, and authentic folklore customs characteristic of different ethnic groups [31]. As a recreational resource the forest territories

are not sufficiently and comprehensively exploited in terms of cultural ecosystem services provisioning. Since the sustainable and alternative tourism is an important diversification element of the region's economy, especially in remote areas, a focus on them could provide added value to the forest economy sector. Moreover, the recreational forests (resorts forests, urban green areas and systems, and peri-urban parks) provide opportunities for short-term recreation and leisure which play an important role for territorial development.

Identifying forest ecosystems as a natural heritage with significant economic potential, rich biodiversity and high social value can be perceived as a real stimulus for implementation of forest therapy and other forest welfare services. This would be considered a prerequisite for increasing tourist flow in the region, bringing higher incomes to the local population and for promoting and enhancing the economic effect of forest areas. The forests in the case-study region provide a good basis for development of different types and forms of tourism, but their potential has been used to different degrees. Despite continuous diversification and expanded territorial scope of supply and demand in recent years, the tourism in the municipality remains highly concentrated in the main touristic centers—Pamporovo and Chepelare resorts and in the town of Smolyan. At the same time the region is characterized by large number of small, highly dispersed tourist places which implies the need for more active interaction between them in order to develop a regional product with a richer set of recreational elements and better chances for market realization. The future of tourism and recreation at municipal level should be based both on the existing traditions in the tourist centers and on the potential of still undeveloped new destinations and services such as forest therapy and other forest welfare services.

By developing and testing an up-to-date topic-based approach for assessing and mapping forest areas in Smolyan, a network of hotspots with "forest therapeutic potential" will be created to directly support the territorial decisions of forest managers. Furthermore, the preservation, promotion and valorization of the local natural, cultural, and historical heritage are key factors for building a sense of belonging and connection of society with nature. Building capacity for self-government, planning, and implementing local policies are crucial for the future development of the remote regions. This corresponds to the Forestry Act of Bulgaria [32], where the public benefits from forests are considered as renumeration for the forest owners after valorization, though without a clear explanation as to how these renumerations will operate. The modification of business processes in tourism is also related with some key factors as global digitalization and personalization of services of tourism industry enterprises [33] which is a challenge for forest managers.

In response to the challenges and opportunities indicated above, the study undertakes an integrated approach to assess and map the potential of forest areas to provide conditions for development of forest therapy services. The methodology includes sequential application of: (i) Selection of a basic spatial unit for evaluation and organization of source database; (ii) Development of a system of criteria and selected indicators and data provision; (iii) Test of the applicability of the indicators and assessment scale; (iv) Integrated assessment of the forest therapy potential; (v) Analysis of the results (Figure 2). The methodological decisions are influenced by the latest studies on the potential of the forest territories to provide public benefits in Bulgaria [2], as well as from the results and discussions in current interdisciplinary research on various applied aspects of Forest Therapy [34–37]. The selection of indicators also considers the methodology for assessing the recreational potential of the territories–Recreational Opportunity Spectrum focused on the importance in reflecting the aesthetics of landscape structure and the territorial accessibility (Proximity) to objects of interest for the research [38].

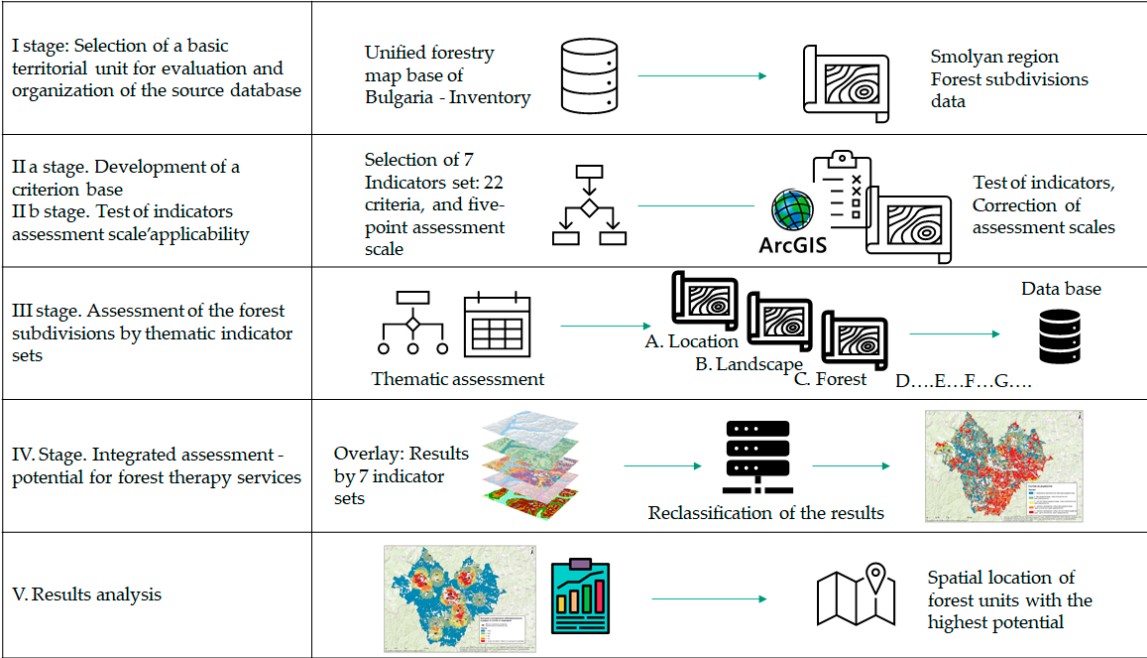

**Figure 2.** Conceptual scheme of the developed up-to-date methodology.

### 2.1. Selection of a Basic Spatial Unit for Evaluation and Organization of Database

Forest subdivisions of forest inventory categorization in Bulgaria are used as main spatial units for evaluation and analysis. Such an approach guarantees particular spatial results and provides opportunities for the preparation of informative cartographic materials. The main source of information is Forest Inventory Database for Smolyan Municipality conducted in 2018 by the Executive Forest Agency—Ministry of Agriculture, Food and Forestry of Bulgaria. The inventory was initiated within the process of preparing a District Development Plan for Forest Territories [31] which is a long-term document on strategic planning of forestry and hunting. This type of regional plan has a pilot character for the Bulgarian practice. Their development is required by the Forestry Act of Bulgaria [32], where "Public Ecosystem Benefits" (actually ecosystem services) are introduced as an argument for forest areas functional zoning.

### 2.2. Development of a Criterion Base

For the purposes of the integrated assessment, the study applies targeted selection of indicators, organized in 7 thematic sets (Table 1). The criteria collection aims to reflect the importance of the basic features of the forest environment in providing conditions for development of forest therapy services: convenient location, favorable landscape features, quality characteristics of forests and waters, terrain and anthropogenic conditions that would contribute to forest therapy facilities development, as well as healing environment. A 5-point scale for assessing the potential for forest therapy is applied to all indicators: from value 1—very low potential, to value 5—very high potential. For selection of the specific indicators, the experience of South Korea was studied. It is the first country in the world which have elaborated and applied a comprehensive policy (including legislation, management structure, facilities, expertise) for providing of forest welfare services [34,38–41]. As understandings of recreation have a strong territorial and especially cultural specificities, the Korean experience has been analyzed against the preferences of Europeans about what they expect from recreation in the forest [42–47]. The development of a criterion base followed tested approaches [48] and available software programmes and databases [49–51]. Last but not least, the indicators are formulated in such a way that the expectations of recreationists can be related to available and accessible database for Bulgaria.

**Table 1.** Criteria and indicators for integrated approach to assess the potential of forest areas for forest therapy services.

| Indicator (Assessment units are the forest subdivisions, Forest Inventory in Bulgaria) | Score | | | | |
|---|---|---|---|---|---|
| | 1 Very low | 2 Low | 3 Medium | 4 High | 5 Very High |
| **A. LOCATION** | | | | | |
| 1. **Remoteness** (from settlements and resorts, minutes) | >80 | 60–80 | 40–60 | 20–40 | <20 |
| 2. **Transport accessibility** (distance from the road network, m) | >400 | 300–400 | 200–300 | 100–200 | <100 |
| 3. **Public transport** (distance from the public transport network, km) | >4 | 2–4 | 1–2 | 0.5–1 | <0.5 |
| **B. LANDSCAPE** | | | | | |
| 4. **Anthropogenic impact** (hemeroby index, degree of naturalness) | Artificial | Far from natural | Semi natural | Close to natural | Natural |
| 5. **View/Panorama** (visibility to other objects, km) | none | ≤20 | 21–23 | 24–26 | ≥26 |
| 6. **Objects causing discomfort within a radius of 2.5 km** (number) | >3 | 3 | 2 | 1 | none |
| 7. **Natural Phenomena within a radius of 2.5 km** (number) | none | 1 | 2 | 3 | >3 |
| **C. FORESTS** | | | | | |
| 8. **Age of the main forest stand** (years) | Non forest area or 0–19 | 20–39 | 40–59 | 60–79 | ≥80 |
| 9. **Composition of the forest stand** (tree species, number and type—coniferous (C) and deciduous (D)) | Non forest area or 1 species | 2 (only C or D) | 3 or more (only C or D) | mixed, 2 species—1 C and 1 D | mixed, 3 or more—both C and D |
| 10. **Density of the forest stand** (classes) | Non forest area or ≤0.3 | 0.4–0.5 | 0.6 | 0.7–0.8 | 0.9–1.0 |
| **D. WATER** | | | | | |
| 11. **Presence of a water bodies** (sea, lake, dam, river, gorge, etc., presence/absence) | none | - | seasonal drying | - | year-round full water |
| 12. **Number of water bodies within a radius of 1 km** | none | - | 1 | - | >1 |
| 13. **Ecological status of surface water bodies** (qualitative assessment, Ecological status and potential, WFD EU) | No water bodies or poor status | - | Moderate | - | Good |
| **E. RECREATION PREREQUISITES** | | | | | |
| 14. **Cultural and historical landmarks within a radius of 2.5 km** (number) | none | 1 | 2 | 3 | >3 |
| 15. **Recreation facilities within a radius of 2.5 km** (number) | none | 1 | 2 | 3 | >3 |
| **F. OPPORTUNITIES FOR DEVELOPMENT OF FACILITIES** | | | | | |
| 16. **Terrain slope** (degree) | >30 | 21–30 | 11–20 | 5–10 | 0–4 |
| 17. **Subdivision area** (ha) | <1 | 1–2 | 2–3 | 3–4 | >4 |
| 18. **Legal restrictions** (function of the subdivision) | Water supply area | Protected area | Hunting territory | - | none |
| 19. **Natural Disaster Hazard** (landslides, fires, floods, risk rate) | High | - | Medium | - | Low |
| **G. HEALING ENVIRONMENT** | | | | | |
| 20. **Altitude** (m) | <100 | <2000 | 100–200 or 1500–2000 | 200–300 or 1000–1500 | 300–1000 |
| 21. **Phytoncides** (ppt/day) | <1 | 1–1.5 | 1.5–2 | 2–2.5 | >2.5 |
| 22. **Anions** (number/cm$^3$) | <150 | 150–170 | 170–1000 | 1000–1500 | >1500 |

The initiation, development, and management of recreational activities in the distinctive conditions of the forest environment requires systematic and consistent information on the preconditions and constraints available for such activities, which goes beyond the thematic scope of the usual forestry database. Such an "extended" inventory, including through cartographic visualizations, provides a basis for adequately informing forest owners of the prospective options, as well as for a coherent planning process—internally (given different forest ownership) and externally (cross-cutting policies, other forms of land use and resource management).

The Recreation Opportunity Spectrum is a proven effective tool for identifying and mapping recreation resources [38]. In this study, the sets of indicators for forest inventory for 'forest therapy services' are considered: Physical Setting, which determines the naturalness of the environment and its spatial remoteness from human influence, its attractiveness and ecological qualities; Social Setting—reflected here by proximity to resort centers and settlements with developed recreational facilities; and Management Setting—considered through factors of transport accessibility, establishment of necessary infrastructure, recreation prerequisites and opportunities. The specific aspects to be assessed and analyzed are explained as follows:

Thematic set A. LOCATION

Important aspects of the spatial location of the forest subdivisions are assessed:

- The remoteness of the respective forest location from settlements and resorts is assessed by estimating the time for access (in minutes). The locations of active sites from the Municipal register of categorized tourist sites and accommodation [49] are entered as initial data. Forest subdivisions with the highest potential are assessed if they are accessible for 20 min or less, and with the lowest potential—those accessible for more than 80 min. The technical implementation was conducted through the development of Model builder transport accessibility [48,50];
- Transport accessibility is assessed through the distance of the subdivision from the road network (m), where the proximity of a road below 100 m is assessed as the most favorable;
- Public transport is estimated through the distance of the subdivision from the public transport network (km). Data from Unified forestry map base of Bulgaria—Inventory 2018 were used, supplemented with available data from Open Street Map.

Thematic set B. LANDSCAPE

This group of indicators reflects distinctive aspects of the landscape structure as an environment for forest therapy:

- Anthropogenic impact is assessed using the degree of naturalness of the landscape performed by reclassification of the land cover types [51] according to the hemeroby index scale;
- Panorama—visibility to other objects (km). The rating scale here is developed in relation to orographic features of the southern part of the Rhodopes Mountain—medium and high mountain relief and deep river valleys;
- Objects causing discomfort within a radius of 2.5 km from the border of the forest subdivision—the indicator detects the presence of objects or visual features of the landscape that are incompatible with the objectives of forest therapy and interfere with its application (mine, excavation, embankment, tailings, depot, logging, landfill, burned area);
- Natural Phenomena within a radius of 2.5 km from the border of the forest subdivision—the concentration of natural heritage sites (caves, waterfalls, springs, and rock phenomena, ancient and primeval trees) is taken into account, where the presence of 4 or more sites is assessed with the highest value. Data from the National Ecological Network are included, supplemented by Open Street Map information.

Thematic set C. FORESTS

It is aimed at reflecting structural characteristics of forest stands which form an environment with a direct beneficial effect on the psycho-physiological comfort of visitors and are a determining prerequisite for implementation of forest therapy:

- Age of forest stand (years).
- Composition of forest stand (tree species, number, and type).
- Density of forest stand (classes).

The mixed forests with three or more tree species, in older age classes, with high density are assessed with the highest value [42–47]. The data for these indicators is available from the Unified Forestry Map Base of Bulgaria—Inventory 2018 [30].

Thematic set D. WATER

It is aimed at reflecting two important aspects related to the presence of water bodies in a forest-dominated environment—quality of the environment with a beneficial impact on human health, and diversity and aesthetics of the landscape, with additional effect on visitors' sensory perceptions. As indicators here are selected:

- Presence of a water body (sea, lake, dam, river, gorge, etc.) in the forest subdivision (presence/absence).
- Number of water bodies within a radius of 1 km (number)—the study uses information from the official database of project "Integrated water management in the Republic of Bulgaria", Japan International Cooperation Agency [52] (for the Ministry of Environment and Water of Bulgaria).
- Ecological status of surface water bodies—qualitative assessments from the ecological monitoring in the documents of River Basin Management Plan 2016–2021 are used (Bulgarian Basin Directorate—East Aegean Region) corresponding to the statements of The EU Water Framework Directive [53].

Thematic set E. RECREATION PREREQUISITES

This group of indicators is aimed at registering basic prerequisites for organization of tourist and recreational activities—sites that testify to the natural and cultural heritage of the territory and often have significance of a tourist destination:

- Cultural and historical landmarks within a radius of 2.5 km (number).
- Recreation facilities within a radius of 2.5 km (number).

The database of National register for cultural heritage (National Institute of Immovable Cultural Heritage, Ministry of Culture of Bulgaria) [54] and Open Street Map [55] is used as input data.

Thematic set F. OPPORTUNITIES FOR DEVELOPMENT OF FACILITIES

This series of criteria includes important conditions for the construction of necessary infrastructure:

- Terrain slope (degree);
- Subdivision area (ha);
- Legal restrictions—functional purpose of the territory and priorities of its management related to nature protection or water protection regimes;
- Natural Disaster Hazard (landslides, fires, floods)—risk level (low, medium and high) in accordance with the national surveys and qualitative classifications. The source collection includes a Unified Forestry Map Base of Bulgaria—Inventory 2018: fire risk [30]; River Basin Management Plan 2016–2021, East Aegean Region: flood risk [56]; Ministry of Regional Development and Public Works, Geoprotection Pernik: Landslide register [57].

Thematic set G. HEALING ENVIRONMENT

This group is essential for the purpose of the study, but the formulation of indicators and rating scales are among the biggest challenges facing it. This is due to the lack of specific quantitative data to assess the indicators. The indicators and values set out are based on South Korea's current recreational forest assessment regulations [34,39,40]:

- Altitude (m)—in terms of its impact on the human body [58,59];

- Phytoncides (ppt/day)—Phytoncides are volatile (rarely non-volatile) fractions of plant essential oils that are capable of killing microbes, pathogenic bacteria, viruses and others. They are "natural antibiotics" and air rich in phytoncides, which has an extremely positive effect on the human immune system. Different tree species release different amounts of phytoncides and on this basis the scale includes a range from 1 to more than 2.5 ppt/day, which are conditionally related to the types of forest plantations;
- Anions (number/cm)—negative ionization of the air also stimulates the human immunity. The scale includes a range from 0 to more 1500 number/cm$^3$, expressed again by an indirect indicator—presence of water bodies, incl. objects with high water dynamics such as waterfalls and fast-flowing waters in high watersheds;

The following data were used for the purposes of this assessment: Copernicus Land Monitoring Service—EU-DEM [60]; Unified Forestry Map Base of Bulgaria—Inventory 2018: Subdivision Type [30]; CORINE land cover database 2018, Bulgaria [51]; Open Street Map—Waterfalls [55]; JICA database, Bulgaria—water bodies [52].

### 2.3. Applicability Test of the Scale for Assessment of the Indicators

The study performed a preliminary test of the proposed criteria and parameters for assessment of the forest subdivisions in the Smolyan Municipality according to the availability of data with the appropriate level of detail and accuracy. This necessitated a revision of ranges of parametric values in the scale for evaluation of some criteria (Remoteness, Transport accessibility, Panorama and Ecological status of water bodies) for correct reflection of the importance of the respective indicator and for objectivity of the spatial assessment. When insufficient volume of input data was established, extended information was sought from publicly available sources such as Open Street Map and BG Mountains online map.

### 2.4. Thematic and Integrated Assessment

The results of the evaluation of the thematic sets of indicators are derived as an arithmetic mean of the scores of the individual indicators in the group. The results are processed as thematic layers in GIS environment. The thematic layers are united by an overlay for integrated evaluation according to all of the criteria applied in the research. The results of the final overlay were experimentally reclassified into a 5-point assessment scale as qualitative categories of potential of the forest subdivisions to provide conditions for forest therapy (Table 2). Scores on the individual criterion groups have equal weight in the overall assessment of potential for "forest therapy". This decision was entirely driven by the diverse nature, though with some limitations (commented below), of the primary or secondary data obtained in the study, and their spatial and temporal representativeness. The study adheres as much as possible to regularly collected forest inventory data and baseline geospatial data for territorial management. This research approach aims to define an appropriate scientific basis for systematization of results in different spatial scales and comparability of assessments at regional and national level.

**Table 2.** Scale for assessing the potential of forest areas for forest therapy.

| Total Score | Potential for Forest Therapy |
|---|---|
| 22–39 | Very low |
| 40–57 | Low |
| 58–74 | Medium |
| 75–92 | High |
| 93–110 | Very High |

The focus here is on the territorial analysis of the complex results obtained. On this basis, the methodology provides for the application of spatial statistics tools (Getis-Ord Gi*, ArcGIS-Spatial Statistics) to highlight the territorial concentration of polygons: hot

spots with potential for forest therapy. Another important perspective of this approach is to establish an appropriate basis for presentation of information to stakeholders and create an informative environment for business initiative.

All of the procedures are performed with reference to the forest subdivision of the traditional forest inventory in Bulgaria as the main information unit for assessment. The software ArcGIS is used for information processing. The results of all of the steps and procedures in the study are organized in a separate database. All of the technical approaches are implemented with the tool ESRI ArcGIS Desktop 10.2 (Spatial Analyst Toolset, Geostatistical Analyst Toolset, Data management Toolset, Conversion Toolset) [61].

## 3. Results

### 3.1. Applied Methodological Approach

#### 3.1.1. Results per Sets of Indicators

The results obtained from the evaluation of the sets of indicators show a mean score of 2.7 (Table 3). It is a clear indication of medium potential of the case-study region for supply of forest therapy services. The highest values (3.3) was distinguished for the thematic set Healing Environment. This result can be explained by the geographical and natural conditions of the analyzed territory: mountainous topography, high average altitude (1320 m), transitional climate, dominant forest environment and high watersheds with deep-seated fast-flowing rivers. The indicator group—Landscape, Forest and Recreation—prerequisites stand out with mean values of 2.9, which is considered as a relatively favorable balance in the indicators selected for evaluation. Smolyan Municipality maintains natural and semi-natural landscapes, has good traditions in forestry and popular destination for mountain tourism. The Location and Facilities development sets of indicators show mean values of 2.8 and 2.7, respectively, which correspond to the mountainous conditions of the terrain and indicates a good potential for further development. The Water set has the lowest mean score of 1.5 which is much lower than the others. This could be explained by the karst terrains in some parts of the territory and requires additional refining and enriching the assessment indicators and the source information.

**Table 3.** Forest therapy potential per sets of indicators.

| Set of Indicators | Assessment Scale, Mean | Very Low | | Low | | Medium | | High | | Very High | |
|---|---|---|---|---|---|---|---|---|---|---|---|
| | | Number of Forest Subdivisions, % | Area of Forest Subdivisions, % | Number of Forest Subdivisions, % | Area of Forest Subdivisions, % | Number of Forest Subdivisions, % | Area of Forest Subdivisions, % | Number of Forest Subdivisions, % | Area of Forest Subdivisions, % | Number of Forest Subdivisions, % | Area of Forest Subdivisions, % |
| A. | 2.8 | 3.5 | 5.4 | 32.3 | 41.6 | 42.5 | 40.6 | 19.7 | 11.7 | 2.0 | % |
| B. | 2.9 | 4.1 | 3.6 | 31.0 | 29.1 | 34.2 | 34.4 | 30.6 | 32.6 | 0.1 | 0.6 |
| C. | 2.9 | 18.0 | 3.3 | 9.8 | 5.9 | 35.3 | 34.7 | 30.9 | 44.8 | 6.0 | 0.4 |
| D. | 1.5 | 79.7 | 59.5 | 1.5 | 1.9 | 0.8 | 1.8 | 17.1 | 34.1 | 0.9 | 11.3 |
| E. | 2.9 | 20.9 | 23.0 | 11.1 | 10.6 | 27.5 | 28.7 | 33.4 | 32.6 | 7.0 | 2.7 |
| F. | 2.7 | 0.6 | 0.1 | 40.5 | 23.1 | 44.3 | 43.8 | 14.6 | 32.9 | 0.0 | 5.1 |
| G | 3.3 | 0.0 | 0.0 | 7.8 | 1.8 | 48.3 | 35.8 | 43.9 | 62.3 | 0.0 | 0.0 |

A. Location, B. Landscape, C. Forest, D. Water, E. Recreation prerequisites, F. Facilities development, G. Healing environment.

The analysis of specific results by indicator groups shows that 42.5% of the total number of polygons have a medium potential for forest therapy, according to the Location criteria (Table 3; Figure 3). This corresponds to 40.6% of the assessed forest area. However, the largest area is occupied by forest subdivisions assessed with low potential for territorial accessibility (41.6%). The forest polygons with the most favorable location in terms of the applied criteria are about 20% of the number of subdivisions distributed over 12% of the total forest area. Considering the mountainous area, poor public transport and poorly developed forest road system, these results are positive for the purposes of the study. The results for the Landscape set show that 1/3 of the territory has a very high potential, which fully corresponds to the expectations for a mountain municipality with a high degree of naturalness of landscapes and representative natural heritage sites. The highest proportion of polygons (34.2%) and the highest proportion of forest area (34.4%) were rated as medium

potential. The Forest set are scored highest in the analysis of the structural characteristics of the polygons, with 44.8% of the area of forest subdivisions having high potential and 11.3% having very high potential. This means that more than 56% of the analyzed area has forest characteristics most conducive to forest therapy. These results are supported by the high values obtained for areas with medium potential. Less than 10% of the forest area falls into the low and very low potential categories.

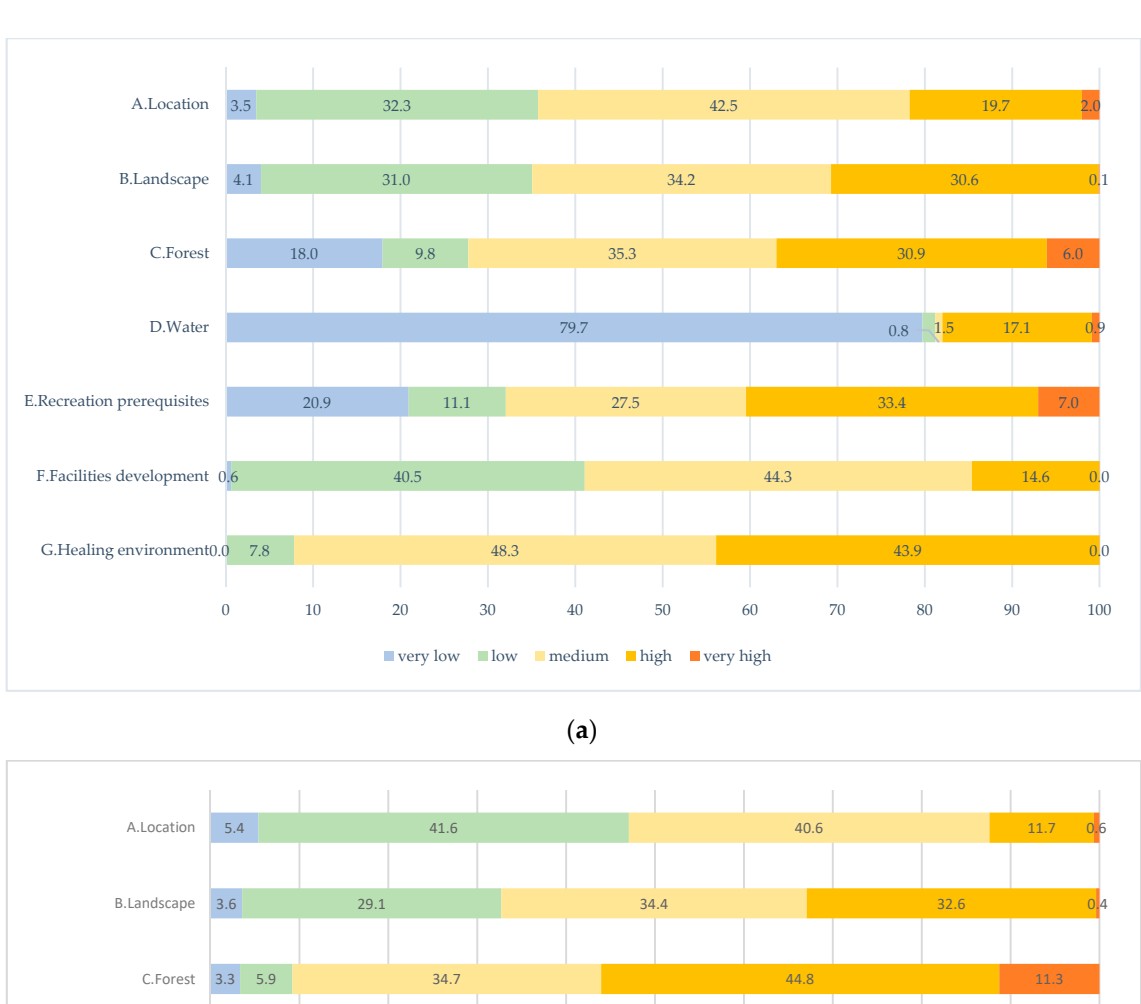

(**a**)

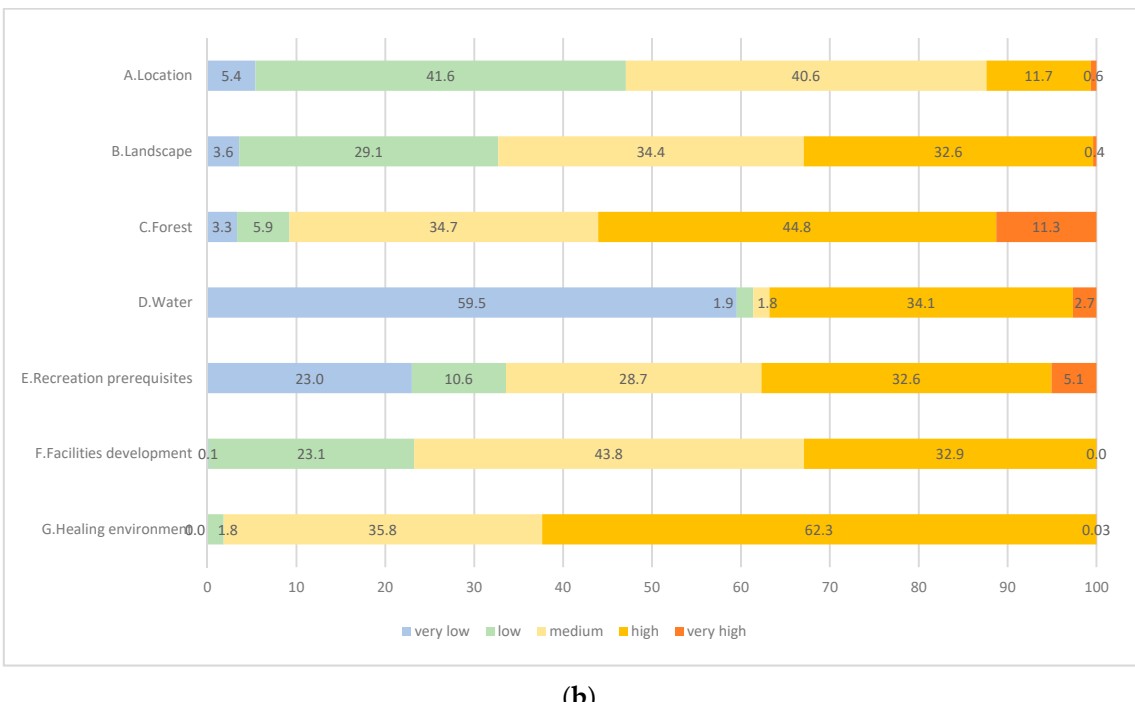

(**b**)

**Figure 3.** Potential for forest therapy by thematic sets of indicators (**a**) Potential for forest therapy by thematic sets of indicators—total number of forest subdivisions, %; (**b**) Potential for forest therapy by thematic sets of indicators—total area of forest subdivisions.

As stated above, the results of indicators in the Water set are a good basis for further analyses—both from a methodological point of view and on technical performance of the assessment. Almost 60% of the area (80% of the total number of polygons) stands out with very low potential and 34% of the areas with high potential (17% of the polygons). These data require an in-depth geographic analysis of the spatial location of forest areas to support or challenge the credibility of the results. There are a lot of karst territories in Smolyan Municipality, which is relevant to the total number of water bodies represented in the territory and included in the study. At the same time, a very important factor influencing the final assessment is the very high number of forest subdivisions (29,579) and their small mean area (2.2 ha). In the vast majority of them there are no water bodies, or they are not full throughout the year, which largely determines the low score of this set of indicators.

A good balance in terms of ratio of number and area of forest subdivisions was found in the results for the Recreation prerequisites criterion (Table 3; Figure 3). The large total share of areas with high (33.4% number of subdivisions and 32.6% area) and medium potential (27.5% number of subdivisions and 28.7% area) gives reason to evaluate the existing circumstances as a good preliminary basis for upgrading the established recreational practices in the region with forest therapy services. At the same time, it should be noted that in 23% of the area, the available preconditions are very limited or absent (very low potential). An in-depth spatial analysis of territorial connectivity of forest areas with high potential is needed, as well as creating of new recreational facilities and expansion of existing ones—circumstances commented below.

Nearly 33% of the study area showed a high potential and 44% showed a medium potential to the Facilities development criteria. The overall results are favorable despite the negligible proportion of subdivisions with very high potential. Approximately 23.1% are sites with low potential. Of all the results commented so far, the highest values were obtained for the Healing environment group—62.3% of the area has high potential (43.9% of the subdivisions). The highest number of subdivisions (48.3%) had medium potential, corresponding to 35.8% of the area.

3.1.2. Territorial Analysis of Results

Considered in a geographic context, the results show significant differences by territory. According to the Location criteria, forest areas with very high and high potential are the contact areas with the settlement structures of Smolyan, Vievo, Petkovo and Momchilovtsi in the central and northeastern part of the municipality (Figure 4a). There is also a clear territorial contiguity with subdivisions of medium potential which is a good territorial prerequisite for the initiation and deployment of forest therapy services in these areas. The western parts of the municipality (around the village of Mulga) are isolated (with very low potential for Location). However, it is these partitions that have a high degree of naturalness—they contain forest polygons with the highest potential in terms of Landscape indicators (Figure 4b). With high a medium potential, the lands adjacent to the settlements of Gela, Solishta, Shiroka Laka, Smolyan and Koshnitsa (western and central parts) stand out. With low and very low potential are south-eastern partitions of the municipality, where a higher degree of anthropogenic impact is observed.

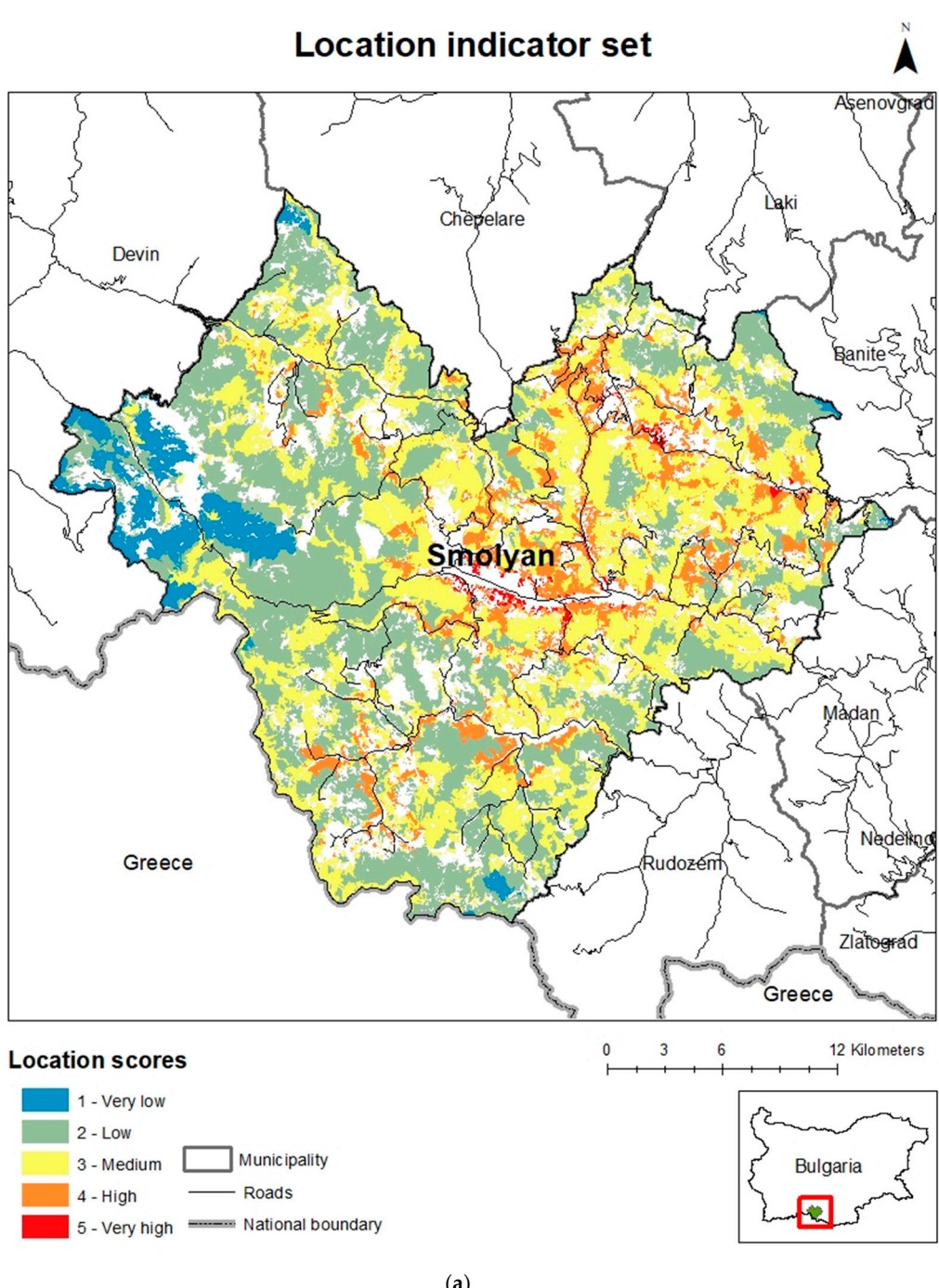

(**a**)

**Figure 4.** *Cont.*

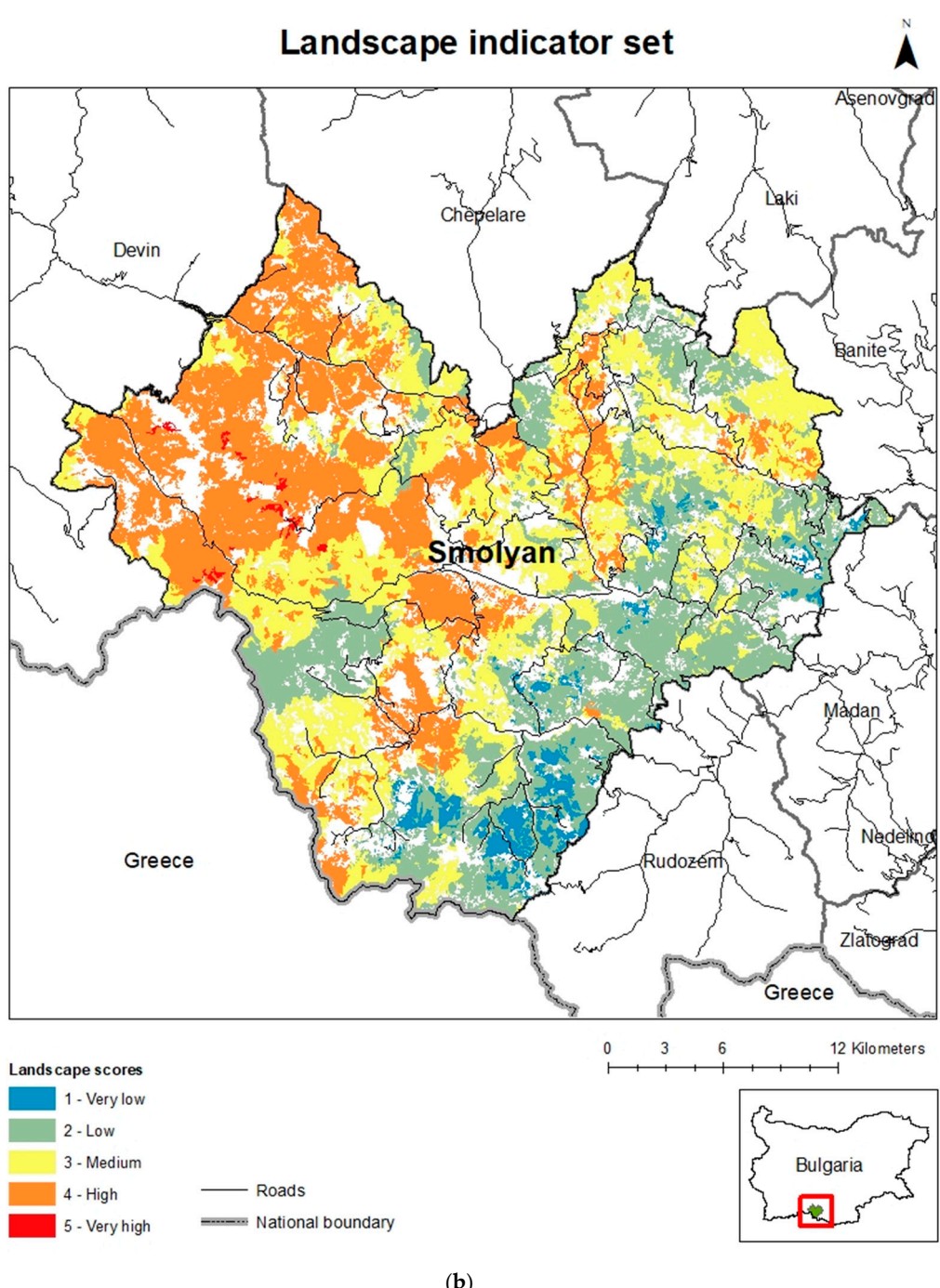

(**b**)

**Figure 4.** *Cont.*

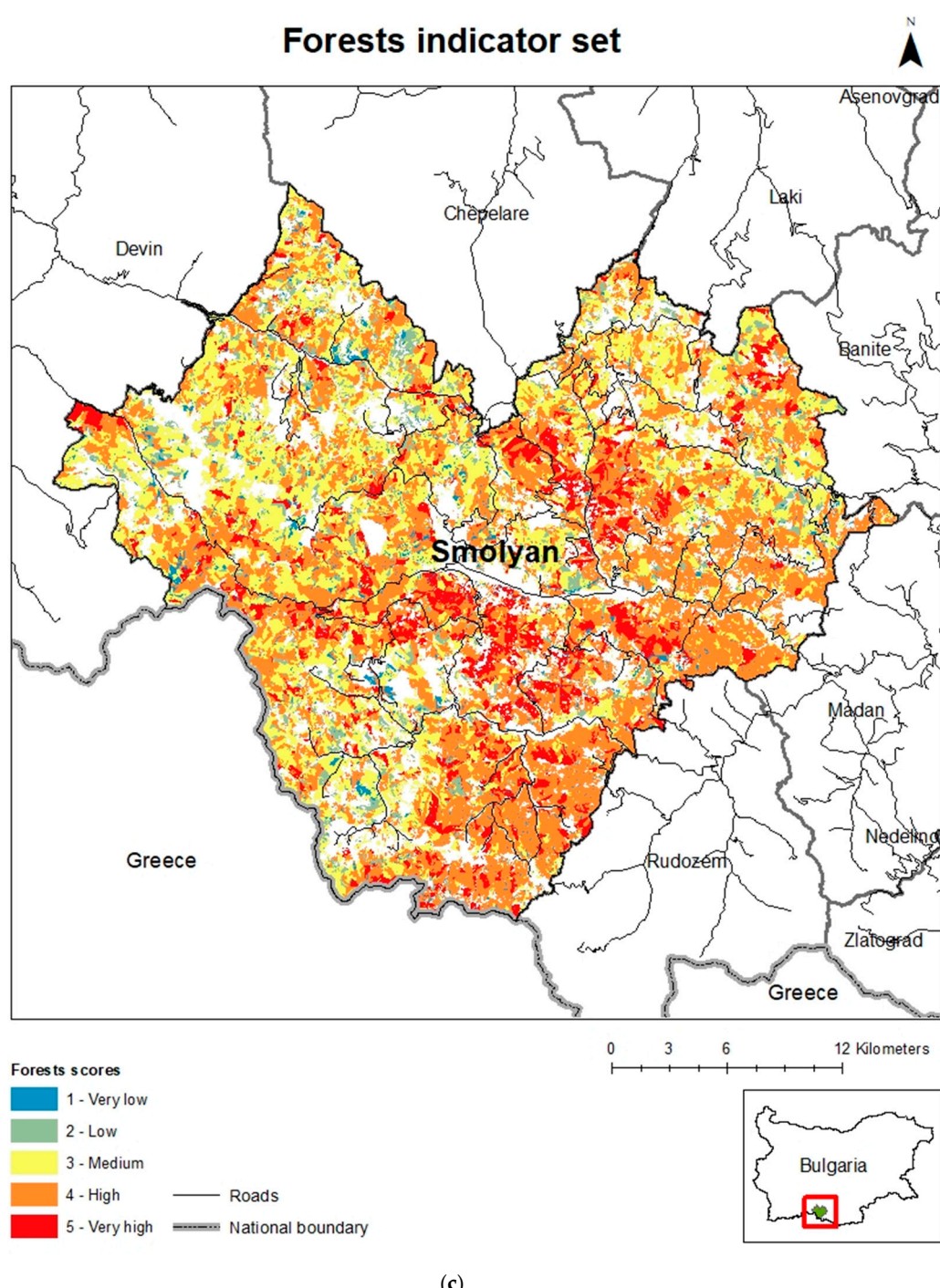

(**c**)

**Figure 4.** *Cont.*

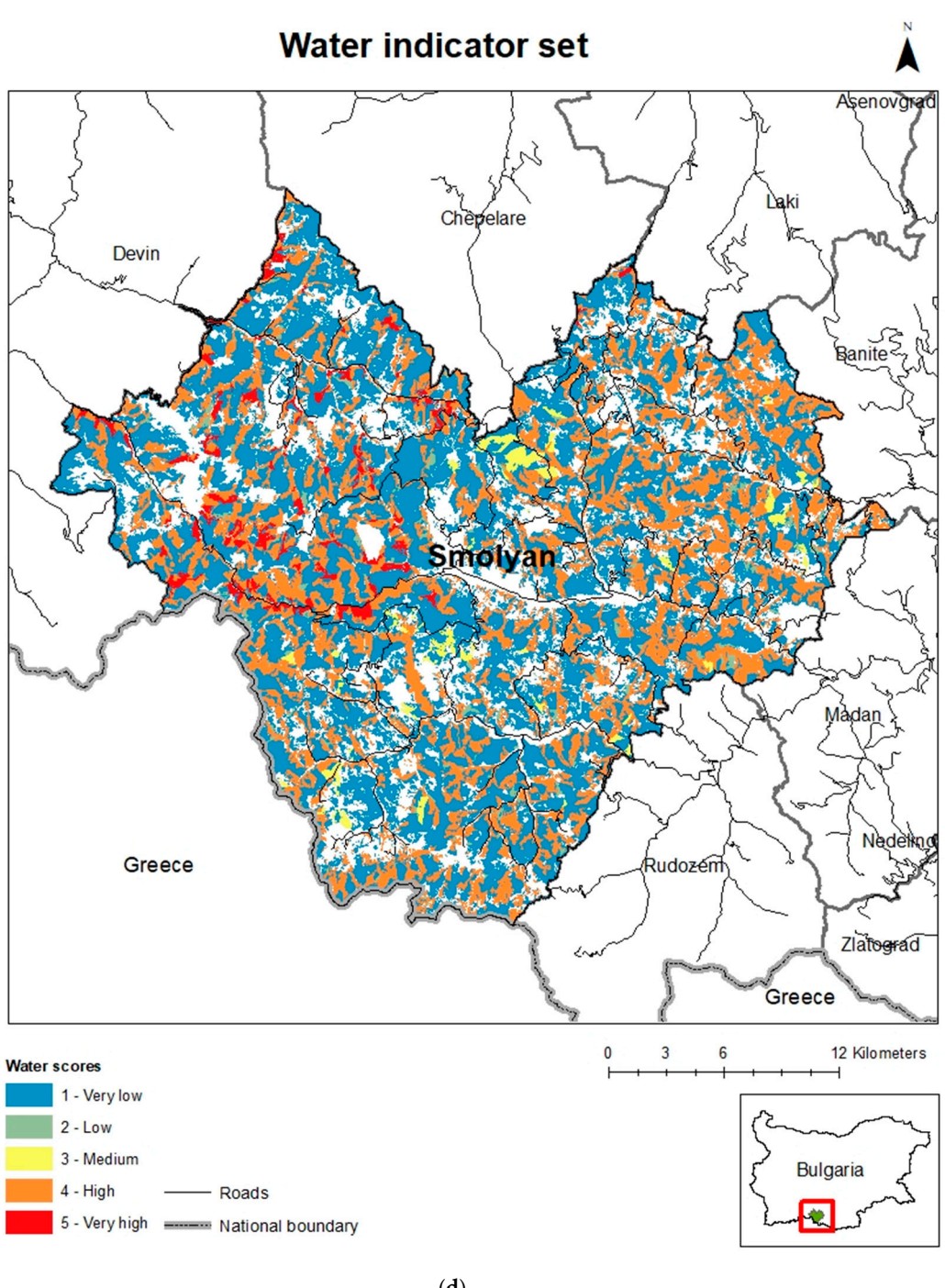

(**d**)

**Figure 4.** *Cont.*

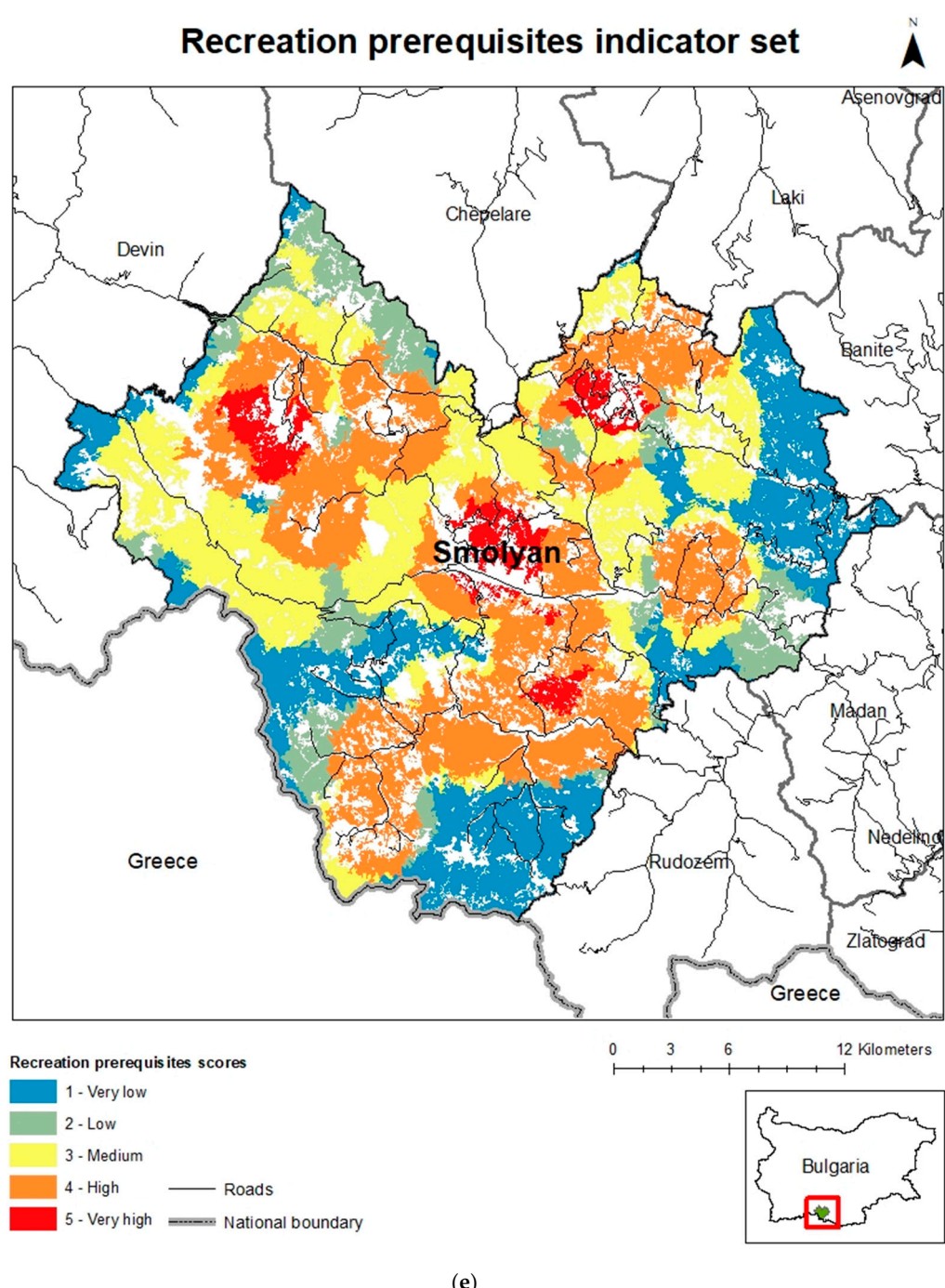

(**e**)

**Figure 4.** *Cont.*

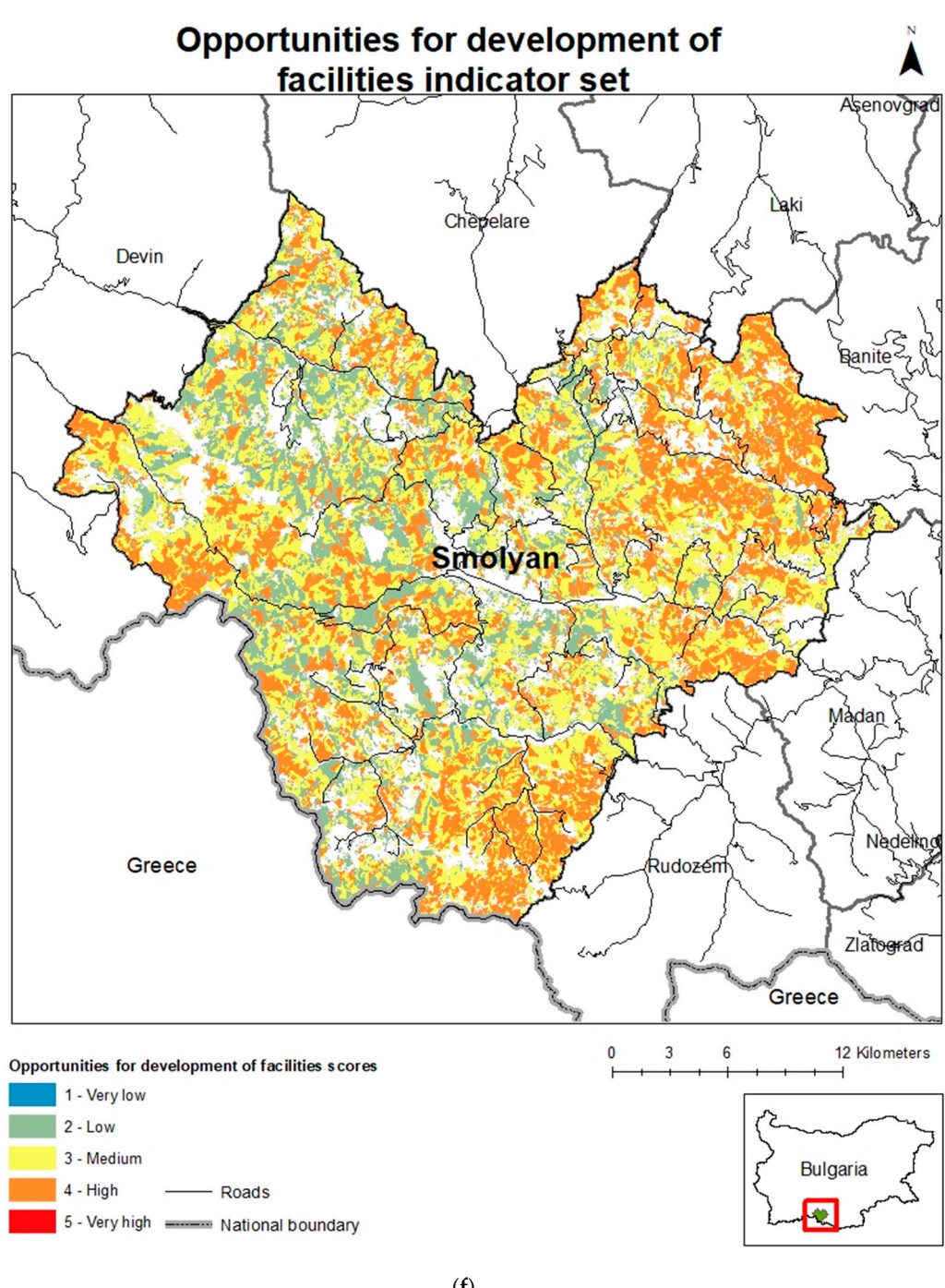

(**f**)

**Figure 4.** *Cont.*

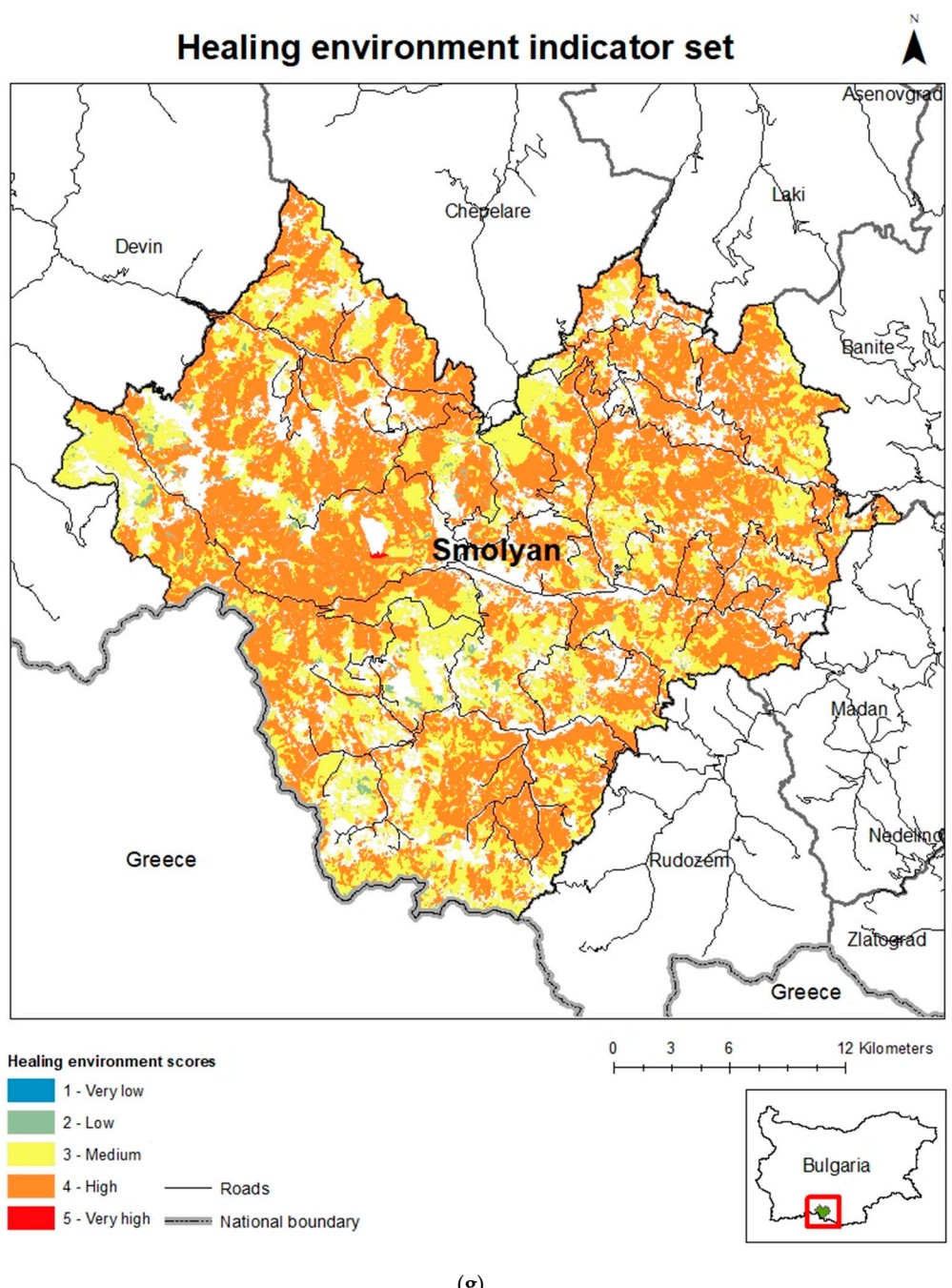

(**g**)

**Figure 4.** Mapping of the thematic sets of indicators ((**a**–**g**) maps).

The visualization of the results in the Forest indicator sets (Figure 4c) shows not only high overall scores for the indicators but also differentiated spatial location of the subdivisions with very high potential and their clear adjacency to those with high potential over a wide area. We defined this result as very favorable for the purposes of the study. Higher spatial densities of such polygons are found in the central and eastern parts of the territory. This is a direct result of the assessment scales applied in terms of forest characteristics and in particular of vegetation diversity. In the eastern part of the territory there is a transition in the natural geographical conditions at the boundary between the high massive Western Rhodope Mountains and the hilly volcanogenic Eastern Rhodopes which affects the diversity of the forest vegetation. The results for the Water criterion show an overall low potential (Figure 4d) but the spatial analysis draws attention to some important circumstances: the high-potential polygons account for 1/3 of all polygons

(Table 3) and they are relatively evenly distributed over the territory and have better areal characteristics than the low-scoring polygons. The forest subdivisions with the highest potential belong to forests in the catchment of rivers Cherna, Mugla and Shirokolashka. Optimal spatial proximity of polygons with high potential and favorable areal extent is universally reported. There is a clear territorial grouping of polygons with different scores in the criterion set Recreation prerequisites (Figure 4e). The forest areas located in the villages of Gela, Smolyan, Momchilovtsi and Chokmanovo have very high potential. The concentration of polygons with high potential is noted in the periphery of the listed mountain localities, as well as in the range of villages Solishta, Koshnitsa, Mogilitsa, Arda, Gradut, Stoykite. Forest subdivisions with medium potential are located in the northern half of the municipality (villages of Mugla, Shiroka Laka, Levochevo). A good territorial balance is noted in the results of the Facilities development criterion set—throughout the municipality there are forest areas with high and medium potential (Figure 4f).

The comparative analysis with the results in the previous group (Recreation prerequisites) provides us with a reason to point out that there are good prospects for development of territories with currently untapped potential. Such territories stand out in the eastern part of the municipality (villages of Bukacite, Gorovo, Laka, Slaveyno and Petkovo) at the contact with municipalities of the Eastern Rhodopes mountains.

The analysis of the spatial distribution of the subdivisions with high and medium potential in the Healing environment criterion group shows an optimal spatial differentiation and favorable overall area characteristics of the polygons (Figure 4g). Areas of very high potential are limited but among them a forest area clearly stands out attached to the Canyon of the Waterfalls site, west of the town of Smolyan.

### 3.1.3. Integrated Assessment

The integrated assessment of forest areas in Smolyan Municipality clearly shows a medium potential for deployment of forest therapy activities—73.7% of the territory (60.4% of the polygons) (Table 4; Figure 5). The total area of subdivisions with high and low potential is similar—respectively 12% and 14%. But there is a big difference in terms of number of polygons—the number of subdivisions with high potential is 1383 (4.7%) but the subdivisions with low potential are 10,225 (34.6%). Polygons with very low potential are only 0.1% of the area (0.4% of the number), and those with very high potential are completely absent.

**Table 4.** Integrated assessment of the forest therapy potential.

| Forest Therapy Potential | | | | | | | | | |
|---|---|---|---|---|---|---|---|---|---|
| Very Low | | Low | | Medium | | High | | Very High | |
| Number % | Area % | Number % | Area % | Number % | Area % | Number % | Area % | Number % | Area % |
| 0.4 | 0.1 | 34.6 | 14.3 | 60.4 | 73.7 | 4.7 | 12.0 | 0.0 | 0.0 |

### 3.2. Contribution of the Individual Indicators

The greatest contributors to the formation of high potential areas for forest therapy are the geographical conditions reflected by indicators 1 (Remoteness), 7 (Natural Phenomena), 11 (Presence of water bodies), 15 (Recreation facilities) and 21 (Phytoncides) (Figure 6; Table 1). They are well supported by the conditions represented by indicators 4 (Naturalness), 6 (Objects causing discomfort), 8 (Age of the forest stand), 20 (Altitude). Circumstances behind indicators 3 (Public transport), 5 (View/Panorama), 16 (Terrain slope) and 19 (Natural Disaster Hazard) received the lowest overall scores and may play role of limiting factors for organization and implementation of forest therapy.

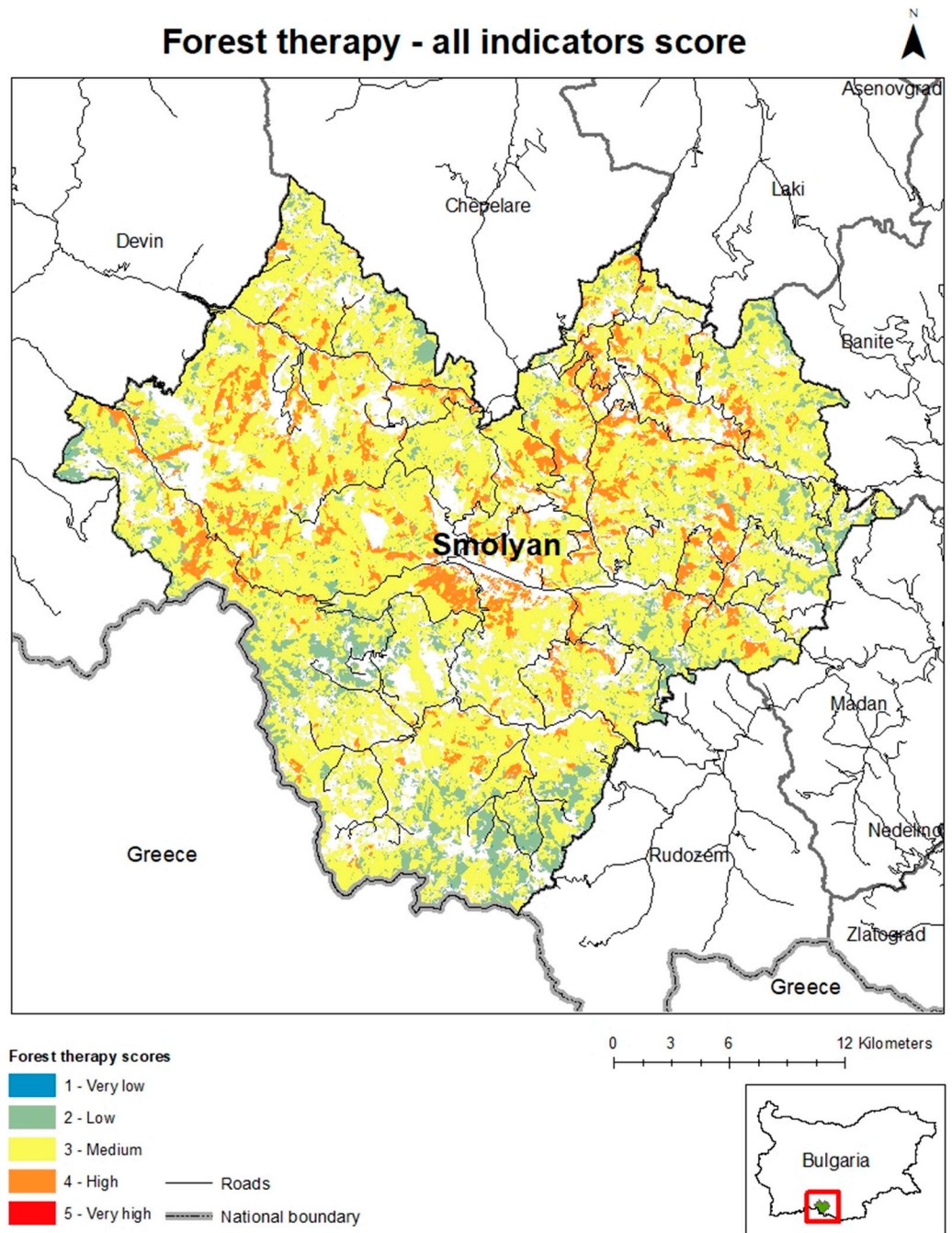

**Figure 5.** Map of the integrated assessment.

The practical orientation of this study and the link to planning forest management and business draw the attention to the need for a precise spatial analysis of data from this type of assessment. The five-point scale offers good preconditions for such an analysis and is appropriate for the methodological framework tested. Additional arguments in this regard are the mountainous character of the case-study region and the small average size of the forest subdivisions used as main spatial information unit for the assessment. On this basis, the contribution of the individual indicators is analyzed for identification of hotspots—territorial concentration of areas with high potential for forest therapy.

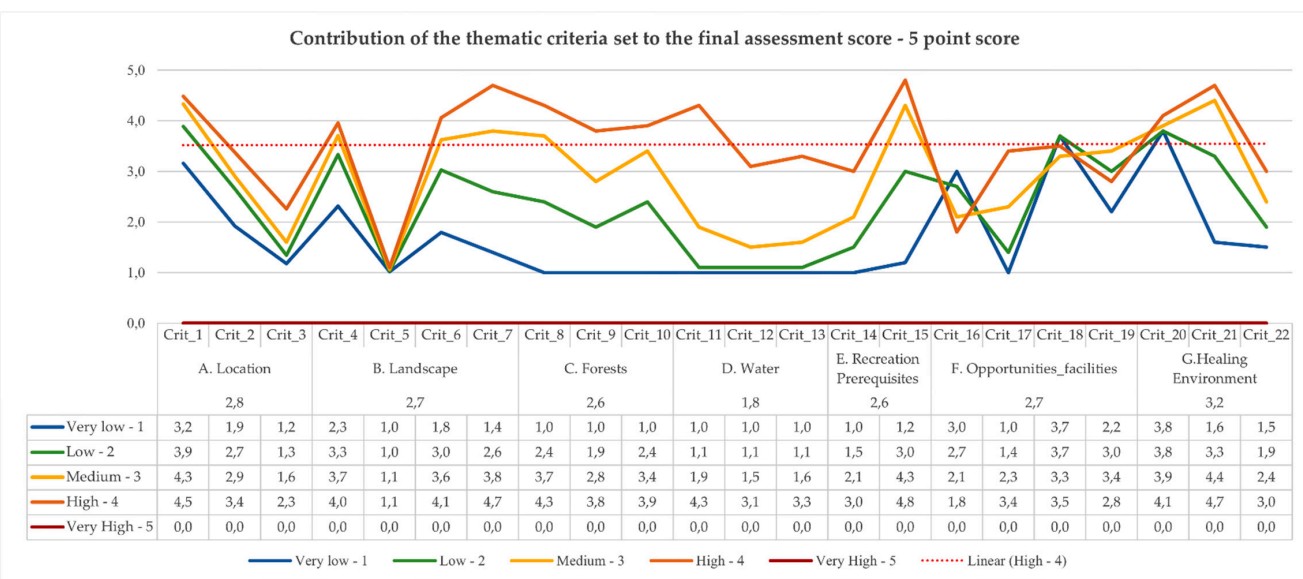

**Figure 6.** Contribution of the individual indicators for formation of high potential for forest therapy.

### 3.3. Territorial Concentration of Polygons with Potential for Forest Therapy—Hot Spot Analysis

In practice, an average area of 150–400 ha is needed to establish a forest therapy center [62,63]. This induces the need to identify concentrations of very high and high potential forest areas bordering on polygons of at least medium potential. Further attention is required to the number of polygons involved in these localities. The applied tool Getis-Ord Gi* (ArcGIS-Spatial Statistics) identifies statistically significant spatial clusters of high values (hot spots) and low values (cold spots). The results provide us with a reason to point out that in the territorial scope of the pilot region there are hot spots territorially attached to the lands of established resort villages Smolyan, Gela and Momchilovtsi (Figure 7a). There are also concentrations more limited in territorial scope which may have independent significance (Mogilitsa and Mugla) or be a territorial link between the hot spots (Bostina and Stoykite). The inclusion of an additional requirement for a shared boundary between polygons demonstrates results that support the generalizations derived above and specify to a high degree the geographical localities conducive to launching forest therapy initiatives (Figure 7b).

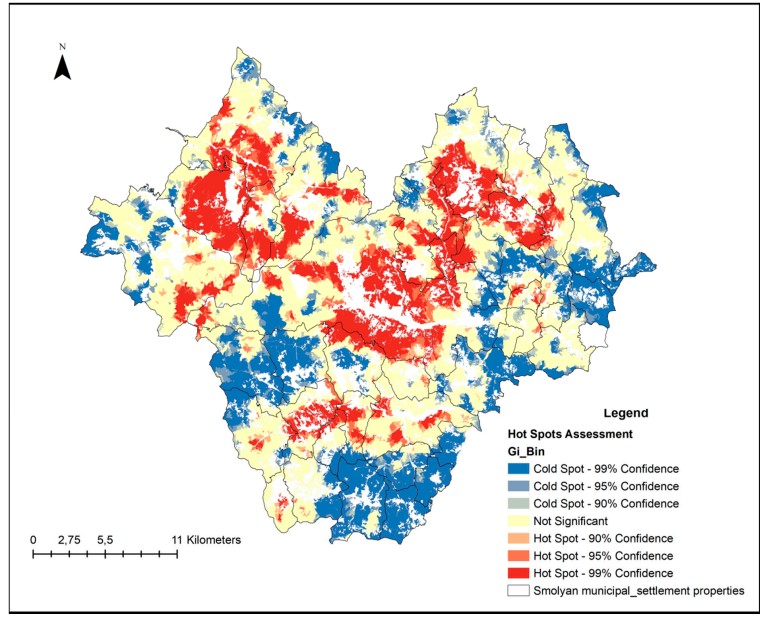

(a)

**Figure 7.** *Cont.*

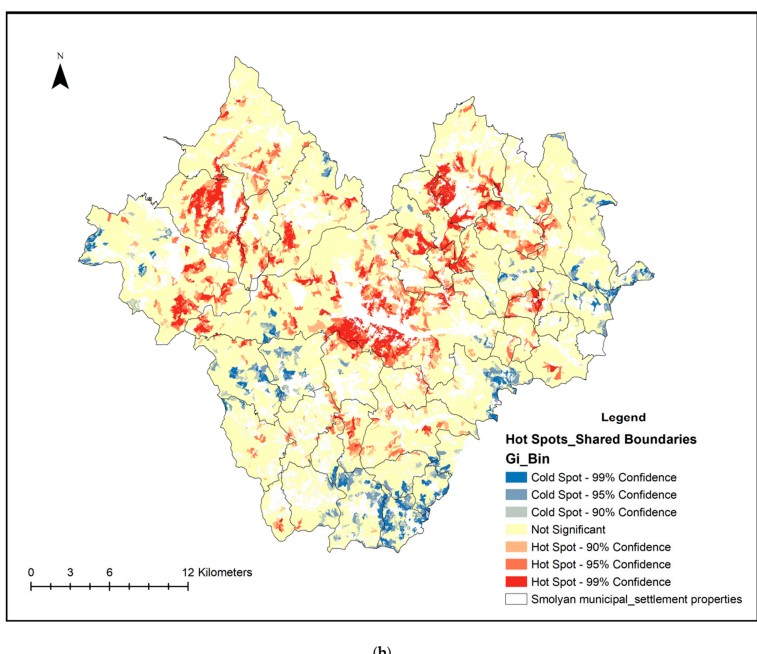

(**b**)

**Figure 7.** Territorial concentration of polygons with high potential for forest therapy (hot spots) (maps (**a**,**b**)).

## 4. Discussion

### 4.1. Methodological Challenges Related to the Indicators for Assessment

The methodology developed in this study uses basic inventory data from forests in Bulgaria to assess the potential for forest therapy services. Since all forests, regardless of ownership type, have been inventoried in Bulgaria, the methodology developed in this study could be an important tool for a broader scale assessment of forest therapy. Furthermore, this makes it applicable at all levels—local (e.g., land in a settlement), regional (e.g., municipality or mountain), and national (for assessment of all forest territories). In addition, the data from the forest inventory are public and freely available on the website of the Executive Forest Agency [64]. Last but not least, they are updated every 10 years, accounting for the changes in forest areas.

However, the use of forest subdivisions also has some disadvantages, which were solved by the methodology proposed through the application of scale for selected indicators for more precise assessment in order the results obtained to be applicable by the forestry managers. The forest subdivisions have a relatively small area (for Smolyan case-study their average area is about 2.2 ha) and often they have no identified permanent borders. Their characteristics are determined on the basis of similar stands in a given area, and in the following inventory their designations and boundaries often change. In the process of defining the evaluation indicators, we encountered a number of challenges and limitations. Two additional evaluation indicators were considered. The first one is "Number of the owners" in the forestry subdivision imposed by the logic that the more owners, the more difficulties for realization of an investment intention for forest therapy. The second is "Existing infrastructure" within the subdivision as the presence of electricity or water supply would greatly facilitate the construction of forest therapy facilities. However, these two indicators were excluded due to the lack of such data at the forestry subdivision level. Another methodological challenge was related to the assessment of the group of indicators "Water". Table 3 shows that its average score is 1.5, which differs significantly (almost twice lower) than the others. It turns out that the forestry subdivision is not the most optimal spatial unit for evaluating these indicators. Figure 3 shows that 80% of the forest stands have very low potential. The case-study has significant water resources; thus these low scores are unlikely to be very representative. The low potential calculated could be explained with gaps in the initial database and lower precision of the evaluation criteria. The presence of water body in or along the boundary of subdivision was assessed.

However, there are very high number of forest subdivisions (29,579) with a very small average area in the case-study region. In the vast majority of them there are no water bodies, or they are not full throughout the year, which largely determines the low score of this set of indicators. Perhaps, the evaluation criterion should be reformulated to assess the subdivision's remoteness from a water body and not its presence within its boundaries. In any case, the results from the study provide a good initial basis for further revision and optimization of indicators.

Another significant methodological challenge was the evaluation of the indicators "Phytoncides" and "Anions". As there is no data on their content at the subdivision level, they were assessed indirectly using information about tree species composition, the presence of fast-flowing water currents, waterfalls, and others. From Table 3, Figures 3 and 4g, ot is evident that the set of indicators "Healing environment" receives a very high average score.

The approach could be considered as suitable as the results are confirmed by a number of other studies and the experience gained by the local population [65–70]. Forests in urban and natural territories are the main zones for tourism and recreation due to the high concentration of negative air ions [71,72]. The negative air ions concentration in suburban forest green areas has been found to be between 1000 and 2000 ions cm$^{-3}$, and that in most urban parks has been found to be between 300 and 1500 ions cm$^{-3}$, whereas this concentration in attached green spaces has been found to be generally below 1000 ions cm$^{-3}$ [73]. Forests in the Smolyan Municipality are used by visitors for the recreation and restoration of their health condition for decades. In 1967 the Ministry of Health issued an order classifying Pamporovo as a climatic resort. The reason is that the climatic characteristics (sunshine, low humidity, movement of air masses and moderate temperatures in the summer) create wonderful conditions for the population for health tourism and sports [74]. In the period of the 60s and 70s of the last century, specialists from the Scientific Institute of Balneology, Physiotherapy and Rehabilitation conducted research on the climatic background of the region of Pamporovo and found favorable opportunities for climate therapy for functional diseases of the nervous system, metabolic syndrome, lung and dermatological pathologies and others. In addition, the municipality of Smolyan is characterized with a high forest cover (76%) dominated by coniferous tree species which leads to a high content of phytoncides in the air. Half a kilogram of phytoncides is released from 1 dka of pine forest in a day, and 3 kg from juniper forest. For this reason, the air in these forests is almost sterile—1 m$^3$ contains only 200–300 bacteria, while in the urban environment of large cities are found tens of thousands of different microorganisms [75]. Individual tree species, shrubs and herbs emit phytoncides not only in different amounts, but also with different healing properties. For example, phytoncides from the fir destroy the causes of whooping cough, dysentery, and typhoid fever, and those of pine—the bacteria that cause tuberculosis, etc. These data confirm our results for a high assessment score associated to the indicator "Healing environment".

*4.2. Integrated Assessment Scale*

In the study, a 5-point scale for the final integrated assessment of forest territories was applied. The scale follows a mathematical approach, as the difference between the minimum and maximum total number of points that a subdivision can gain from all indicators is divided into five equal parts (20% for each) to obtain 5 final scores. It could be expected that this will result in a very detailed classification, but in practice the area of the case-study region is divided only into three groups—almost 74% of the area has medium potential, 14% has low potential, and 12% has high potential (Table 4). In practice, it turns out that there are no areas with very low and very high potential, but this is a logical and explicable result as the municipality of Smolyan is a typical mountainous region with beautiful natural and climatic conditions and relatively developed tourist infrastructure, which explains very small percentage (0.1%) of the areas with very low potential. However, the methodology has been developed in order to be applicable for

the whole country; in the plain and sparsely afforested territories the percentage of the areas with very low potential will be much higher. As for the lack of areas with very high potential, the reason is in fact that most of the indicators depend on the investments in the forest territories. Figure 6 shows that the indicators with the lowest scores are 3 (Public transport), 5 (View/Panorama), 16 (Terrain slope) and 19 (Natural Disaster Hazard). Their assessment scores can vary and can be increased through well-defined and target investments on the locations, where the therapy complexes will be established, for example: the optimization of public transport through more stations and lines, especially to serve for these locations; the establishment of viewpoints on the identified sites and realization of all related activities as cutting the vegetation to clear the panorama and/or to build different facilities (observation towers, lifts etc.); terrain slope can be leveled by excavation and embankment works where infrastructure is needed, and by building eco-paths and leveled sites for visitors to the rest of the territory; Natural Disaster Hazard risk reduction by building drainages, firebreaks, strengthening terrains and riverbeds, etc. It is important that there are opportunities to improve these indicators in selected areas and thus to obtain places with a very high final score. Very often natural phenomena are difficult to explain and evaluate with a purely mathematical approach. This is valid for recreation and therapy, which depend on many factors such as natural resources, infrastructure, cultural features, personal perception etc. Therefore, in the assessment it is necessary to apply flexible approach not based solely on the mathematical result but also on results of surveys. The Republic of Korea is the first country in the world to implement a comprehensive policy for the development of forest therapy and healing, including existing regulations to assess the potential of forest territories [6,76]. In their practice the assessed areas are divided into two groups—suitable and unsuitable for forest therapy. The logic is that in addition to the purely natural resources, the potential for forest therapy largely depends on the subsequent investments that can be made in infrastructure, services, experts, etc., which will further increase the potential of the territory. Areas which gain more than 2/3 of the maximum total number of points are defined as suitable. In the case-study, these are the forests with over 73 points, which correspond to groups with high and very high potential. In this regard, 7,873 ha (12% of the total area) are assessed as suitable for forest therapy in the municipality of Smolyan (Table 4). They can be united in several "hotspots"—around Smolyan, Gela, Momchilovtsi, Mogilitsa, Mugla, Bostina and Stoykite (Figure 7a). Smolyan and Momchilovtsi are popular tourist destinations, so the development of forest therapy in these areas will further diversify the offered services. The other settlements are not popular resorts, but our results show that they are suitable for the construction of forest therapy centers, which will enrich the tourist destinations/routes in the municipality and will create additional jobs. In comparison across Korea, there are about 50 "healing forests" [77], therefore the identified 7 suitable sites in Smolyan Municipality could be considered as a positive and reliable result. It is very important to note that the data obtained must be verified by a field survey in order to make a final decision on the construction of a forest therapy center and to select its optimal location. In order to provide conditions for practicing forest therapy, the hotspots could be in the focus for further construction of specific infrastructure—therapy complexes with accommodation and food facilities, specialized (medical) equipment, therapeutic indoors and outdoors facilities (eco-trails), bathing places, facilities for therapy, recreation, games, leisure, sports, etc.) and additional accompanying infrastructure. Furthermore, for the functioning of such forest therapy centers, in addition to the usual administrative and service staff, special trained experts will be needed ("forest therapy instructors") which is directly linked to the development of innovative and interdisciplinary educational programs and strategies. They will perform an initial general health assessment of visitors through a medical examination and a psychological survey and on this basis, they will develop and offer an optimal individual forest therapy program.

*4.3. Informativeness of the Assessment*

The results of the study would be useful for a wide range of stakeholders as the methodology has potential to provide a new direction in forest management decision making by providing a new low-cost approach for data analysis. Most of the Bulgarian forests are owned by the state ad (70%) and the state has identified recreation as one of the priorities or their management. Therefore, the methodology can directly contribute to the forest managers through indicating which areas in the country are most suitable for providing forest therapy. Moreover, the state has determined 225,000 ha of forests for recreation, including forests near resorts and touristic sites, green infrastructure in urban areas, and peri-urban parks. Currently, no services from these forests are provided to the population [78] and this methodology could indicate which of them have potential for forest therapy. The methodology will also be useful for all other forest owners (municipalities, individuals, public organizations, and others) to diversify the services they provide. This is a prerequisite and a good basis for the emergence and development of various business and educational initiatives which in turn will contribute to sustainable management of forests and improved qualification of various experts and staff involved, especially in remote rural and mountainous areas. The forest therapy is not only of interest to the forest sector but is a promising prerequisite for building cross-sectoral cooperation. Forest therapy has the potential to contribute to tackling various and very topical social and demographic problems [6], so the results of the study will be of great help in planning partnerships between forest, tourist, health, and educational institutions for provision of social services to improve the health and well-being of the population.

Mapping the potential for forest therapy is a digital result that is easily accessible and understandable to the general public. It will benefit tourist organizations and resorts to diversify the services they offer. It will also benefit private entrepreneurs by giving them a basis to plan a new business initiative. Moreover, it will be useful for ordinary tourists when planning their routes, in order to get the most out of the interaction with forest. Last but not least, this is a result with great added value in terms of digitalization of the natural heritage of Bulgaria because the forest ecosystems are an integral part of the country's natural heritage.

## 5. Conclusions

The developed methodology for assessing and mapping the potential of territories for forest therapy integrates the available knowledge on ecosystem services with quantitative and qualitative data from forests and presents a practical tool for foresters and decision-makers in management, territorial planning, and development. The methodology was successfully tested for the case-study area of Smolyan Municipality and the results obtained are reliable and logical, which provides a good basis for both implementation and further precision. The research is a prerequisite for the development of a regional touristic product with a richer set of recreational elements and better chances for market realization, including the potential of still undeveloped new destinations and services.

The methodology is applicable for the other forest territories in Bulgaria and the regions with similar data extracted from forests, regardless of the type and ownership of forests. Moreover, the approach of combining traditional forest inventory data with up-to-date methods for integrated assessing and mapping cultural ecosystem services could be applied in other countries after adapting to their national cultural characteristics and databases available there.

It is a cost-effective and informative methodological approach to support decision-makers, owners and investors in the forestry and tourist business. The results obtained provide a promising fundamental basis for further diversification of these sectors and to attract the attention of management authorities for sustainable territorial development and improvement of human wellbeing. Moreover, forest owners could be supported in choosing new strategic solutions for their property management. The proposed approach

could further help different stakeholders and the general public in meeting the challenges of more efficient, thus benefitting human-forest interaction.

**Author Contributions:** Conceptualization, Y.D., M.Z. and B.B.; methodology, Y.D. and B.B.; software, I.I., L.S., S.N. and S.D.; validation, S.N. and B.B.; investigation, M.Z., M.G. and B.B.; resources, Y.D., M.Z., M.G., S.N., B.B., M.N. and W.-S.S.; writing—original draft preparation, Y.D. and B.B.; writing—review and editing, M.G., B.B. and M.Z.; visualization, L.S., I.I., S.D., S.N. and B.B.; supervision, Y.D.; project management, M.N. All authors have read and agreed to the published version of the manuscript.

**Funding:** This publication is the result of a survey conducted within the project BG05M2OP001-1.001-0001 Establishment and Development of "Heritage BG" Centre of Excellence (Operational Program "Science and Education for Intelligent Growth", priority Axis 1 "Research and technological development").

**Institutional Review Board Statement:** Not applicable.

**Informed Consent Statement:** Not applicable.

**Data Availability Statement:** (1) Corine Land Cover 2018 (vector)—version 20, June 2019. (Source: https://sdi.eea.europa.eu/catalogue/srv/api/records/53ef1493-e7a1-4216-b043-87a7c2a5a68d). (2) Lakes and Rivers in Bulgaria. (Source: The study on integrated water management in the republic of Bulgaria—created for Ministry of Environment and Water by Japan International Cooperation Agency (JICA)). (3) Bulgaria latest free open street map data. (Source: http://download.geofabrik.de/europe/bulgaria.html). (4) Forestry data. (Source: Unified forestry map base of Bulgaria—Inventory (2018), www.iag.bg). (5) Data for the ecological status of surface water bodies (Source: River Basin Management Plan 2016–2021, East Aegean Region). (6) National register for cultural heritage. (Source: http://ninkn.bg/Documents/categoryPreview/13). (7) Ministry of Regional Development and Public Works, Geoprotection Pernik: Landslide register. (Source: http://geozashtita.bg/). (8) Digital Elevation Model. (Source: https://land.copernicus.eu/). All accessed on 22 July 2021.

**Conflicts of Interest:** The authors declare no conflict of interest.

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
