# Peer review of "An Integrated Approach to Assess the Potential of Forest Areas for Therapy Services"

_land, doi:10.3390/land10121354_

Round 1

Reviewer 1 Report

MAJOR
1. Methodology and results are mixed and not well divided. This prevent the reader’s straight forward understanding. Simply, 3.1 should be moved to the methodology part.
2. The key point of this work is the selection, definition of the criteria, and the integration method. the authors explained the selection and criteria of the indicators were strongly depends on the Korean experience (3.1.2), and the score integration was just an arithmetic mean (3.1.4)=very simple scalarization. I think we cannot find novelty of this paper without explanation. The authors need to clearly define the newness and the innovative point of this research.
3. And the selection and criteria of the indicators are critical to the result, but a few explanation in this paper itself. Please add the extra-description and justify the settings.
4. Overall, the description of the ‘innovative’ aspect  of this analytical package looks weak, so I would like to propose the authors to enhance the explanation.

MINOR
1. The abstract should include the summary of the methodology. The current abstract show few model information, not easy to imagine the contents of this paper from the abstract.  

Reviewer 2 Report

Review SUS

 Integrated methodological approach for assessment and mapping of potential of forest areas to provide forest therapy services

Line 34: “ providing services and investments in remote areas” Please provide examples.

Line 36: “ Maintaining and enhancing these functions should be an integral 36 part of the forest policy as» How? This is not clear to me.

Line 39: “ while the intangible benefits of forests for human health and wellbeing are often more difficult to be 40 measured.” This comes very abruptly. The topic could be better introduced.

Line 44ff: “

 Bulgarian forests and» Why Bulgaria? This comes out of the blue.

Line 46ff: “ The economic, social and ecological functions of forests are of significant importance to sustainable development of society and for improving the quality of life,” Maybe you could specify more these functions for your case.

Line 58: “forest provisioning, supporting and regulation & maintenance ecosystem ser-58 vices is statutory and management structures are familiar with them» This should be better introduced. At the beginning you just talk about these services more generally and here you name these more concretely. It would be good to introduce this from the beginning.

Line 62: “Forest Act, National Forest Strategy, a number of regulations and guidelines” Could you give some examples? Be more precise?

Line 66: “comprehensive concept for Forest 66 Welfare Services was recognized and recently its development was initiated” What does this concept mean? Where does it come from? This could be better introduced.

Line 65: “have been developed on how benefits of forest cultural services to reach to a 65 large part of the population» This sentence needs to be revised.

Line 66: 2Welfare Services was recognized and recently” You are using several terms for similar things, or maybe I misunderstood. Check for consistency.

Introduction: In my point of view a research question is missing as well as a clear objective. What I also miss is something about forest owners and monetary activities. Could you also provide the article that allows such activities in the forest? For instance in Switzerland we have an article that gives access to third parties to recreate in the forest, but activities were third parties want to e payed in a grey zone.

Line 95: “The methodology is tested» Which methodology?

Line 96: “Forest lands occupy 65 650 ha in the municipality which represents 76% of its total area” Who is the owner?

Line 100: “well as for development of 100 hunting farms, collection of wild fruits, mushrooms, herbs, and others, but also with their 101 protective and recreational functions.» It would be good for the reader to divide this depending on the ES the forest is offering.

Section Material and methods needs to be revised. Subheadings could help structure more the information you are providing. I totally miss the methods that you used to collect data. There are several parts and are not in my eyes part of this chapter. It is also very long.

Line 151 ff: “The methodology includes sequen-151 tial application of: i) Selection of a basic spatial unit for evaluation and organization of 152 source database; ii) Development of a system of criteria and selected indicators and data 153 provision; iii) Test of the applicability of the indicators and assessment scale; iv) Assess-154 ment of the forests by thematic indicator sets; vi) Integrated assessment of the forest ther-155 apy potential; vii) Analysis of the results» You should expand on that. This is the core of this chapter.

Figure2: Quality is very bad to read. Source are missing in the Graph. All the steps presented in the Graph should be presented more explicitly in the text.

Line 189: “For the purposes of the integrated assessment, the study applies targeted selection of 180 indicators, organized in 7 thematic sets” Is this not part of the methods section?

Results section mixes methodology and results. It must be clearly what is really what you found out. Additionally, it is not clear how the criteria were selected. These seem to come out of the blue.

Source of Table 1 is missing.

It is not clear how the regions were chosen.

Round 2

Reviewer 1 Report

I think the manuscript had been modified sufficiently. 

Author Response

Thank you very much.